# Cryogenic multiplexing using selective area grown nanowires

Dāgs Olšteins [1,2], Gunjan Nagda [1], Damon J. Carrad [2], Daria V. Beznasyuk [2], Christian E. N. Petersen [2], Sara Martí-Sánchez[3], Jordi Arbiol [3,4] & Thomas S. Jespersen [1,2] ✉

Bottom-up grown nanomaterials play an integral role in the development of quantum technologies but are often challenging to characterise on large scales. Here, we harness selective area growth of semiconductor nanowires to demonstrate large-scale integrated circuits and characterisation of large numbers of quantum devices. The circuit consisted of 512 quantum devices embedded within multiplexer/demultiplexer pairs, incorporating thousands of interconnected selective area growth nanowires operating under deep cryogenic conditions. Multiplexers enable a range of new strategies in quantum device research and scaling by increasing the device count while limiting the number of connections between room-temperature control electronics and the cryogenic samples. As an example of this potential we perform a statistical characterization of large arrays of identical quantum dots thus establishing the feasibility of applying cross-bar gating strategies for efficient scaling of future selective area growth quantum circuits. More broadly, the ability to systematically characterise large numbers of devices provides new levels of statistical certainty to materials/device development.

Quantum electronics are rapidly maturing towards large-scale integrated (LSI) circuits incorporating a multitude of interacting quantum devices. There is therefore an onus on potential quantum materials candidates to exhibit both high-precision reproducibility and scalability potential. Semiconductor nanowires (NWs) constitute an important platform for quantum electronics since the electronic confinement intrinsic to the structure simplifies the fabrication of complex devices[1–4], the flexibility of contact materials enables hybridization with important quantum materials such as superconductors[1,3,5–9], and increased capacity for strain relaxation over bulk materials enables exploration of exotic heterostructures[2,6,8,10]. Conventional NWs grown perpendicular to the substrate have been highly successful for high-performance electronics, simple complementary circuits, and in fundamental mesoscopic physics[1–11]. However, the difficulty in fabricating individual vertical devices, and the insufficient precision and yield of techniques for transferring NWs to the planar geometry compatible

with standard semiconductor processing[12–14] has thus far inhibited the development of LSI NW circuits. A promising alternative is the bottom-up growth of in-plane semiconductor NWs directly on a suitable device substrate using selective area growth[15–18] (SAG). In SAG, the positions and dimensions of NWs are controlled by lithographically defining openings in a dielectric mask, enabling the controlled growth of large-scale networks and NW arrays[16,19,20]. While proof-of-principle single NW devices, e.g., field effect transistors (FETs)[21,22], Hall crosses[23], quantum interferometers[20,24], hybrid superconducting devices[24,25], and quantum dots (QDs)[26] have been reported, scalability towards integrated quantum circuits—a central motivation behind the development of SAG and similar bottom-up grown planar nanostructures[27,28]—has not been addressed.

Here, we make the first demonstration of LSI circuits based on SAG. Starting from large arrays of thousands of SAG NWs we fabricate multiplexer (MUX) circuits that operate at the deep cryogenic

[1]Center For Quantum Devices, Niels Bohr Institute, University of Copenhagen, 2100 Copenhagen, Denmark. [2]Department of Energy Conversion and Storage, Technical University of Denmark, 2800 Kongens Lyngby, Denmark. [3]Catalan Institute of Nanoscience and Nanotechnology (ICN2), CSIC and BIST, Campus UAB, Bellaterra, Barcelona, Catalonia, Spain. [4]ICREA, Passeig de Lluís Companys 23, 08010 Barcelona, Catalonia, Spain. ✉e-mail: tsaje@dtu.dk

conditions relevant to quantum electronics. Cryogenic multiplexers are key ingredients towards scaling quantum electronics[29–32] as the number of addressable devices scales exponentially—rather than linearly—with the number of connecting control lines. This is crucial for reducing heat load from wiring between the cryogenic sample and room temperature, and integrated MUX circuits allow highly dense packing of devices utilizing chip-area conventionally required for bonding and routing. Our setup allows us to address and measure 512 individual SAG quantum devices using only 37 control lines.

Our architecture also includes de-multiplexers (d-MUX) connected back-to-back with the corresponding MUX, enabling us to unambiguously confirm the functionality of the circuit, identify faulty operation among the thousands of NW FETs, and self-correct against most failure modes.

Introducing on-chip multiplexing to bottom-up grown nanostructures enables new strategies in quantum electronics research, such as automated searches through large ensembles of devices for rare or exotic phenomena, and systematic, statistically significant exploration of the correlation between device performance and, e.g., materials properties or device geometry. To demonstrate the potential of the latter we perform a statistical characterization of device reproducibility within a large array of nominally identical SAG NW QDs. QD arrays are promising candidates for implementing quantum computation and simulation[33–35], and quantifying device-to-device reproducibility—enabled by the MUX circuit—is crucial for the successful development of cross-bar gate architectures which constitute an important strategy for limiting gate counts in realistic large-scale implementations[36,37]. We find that all QDs of the array can be concurrently tuned to the Coulomb Blockade using only three shared cross-bar gates further confirming the potential of SAG as a scalable platform for quantum devices.

## Results
### Material and electrical properties
Our circuits are based on $[0\bar{1}1]$ oriented InAs SAG NWs grown using molecular beam epitaxy (MBE) on GaAs $(3\,1\,1)$A substrates. See Methods and Supplementary Section S1 for details. Figure 1a shows a cross-sectional high-angle annular dark-field scanning transmission electron microscope (HAADF STEM) micrograph of a single NW, and Fig. 1b shows a combined schematic and atomic force microscope

(AFM) micrograph of an NW section. The conducting InAs channel sits atop an insulating GaAs substrate and GaAs(Sb) buffer[20]. The NWs are terminated with {111}A facets as a consequence of the $(3\,1\,1)$A substrate symmetry, producing the asymmetric cross-section. Detailed structural analysis is presented in Supplementary Section S2. Figure 1c, d illustrate the capacity for scale-up inherent to SAG, through an AFM micrograph and dark-field optical microscope micrograph of representative sections of a $512 \times 16$ array of nominally identical 10 μm-long NWs. The inset to Fig. 1d shows a photograph of a cleaved $5 \times 5$ mm piece of the growth wafer containing ~18,000 SAG NWs, 9216 of which were used for device fabrication; the diffraction from the large arrays is visible.

The Fig. 2a inset shows a scanning electron microscope (SEM) micrograph of a typical device along with a schematic cross-section. The device includes 4 SAG NWs connected in parallel by Ti/Au ohmic contacts and a 1 μm long gated segment (see Methods for fabrication details). Two gates are seen: one which acts on the exposed InAs NWs (blue), thereby controlling the conductivity, and one which is screened by the metal contact, and thus has no effect on the underlying NW (gray). This gate is, however, important for the MUX operation as discussed below. Figure 2a shows the conductance, $G$, as a function of gate voltage, $V_G$, measured at a temperature of $T = 20$ mK for 6 different devices with varying numbers of NWs, $M = 1, 4, 16, 32, 64$ after subtraction of a constant series resistance $R_S$ (see Methods). The devices act as normally-on, $n$-type FETs with identical threshold voltages, and the dashed lines show a common fit to the relation $G = KM(V_G - V_{TH})$ with fixed parameters $K = 0.12$ mS/V and $V_{TH} = -0.4$ V. Except for $M = 64$ where $G$ is somewhat lower than expected, $G$ is proportional to $M$ as expected for equally contributing NWs and the linear scaling with $V_G$ is typical for NW FETs. The deviation for $M = 64$ may be due to a high sensitivity to the estimate of $R_S$ when the device resistance is low (Methods). Importantly, Figure 2a shows that SAG devices manufactured in parallel exhibit consistent $G$ vs. $V_G$, with reproducible, $M$-independent $V_{TH}$, enabling the use of large-$M$ FETs as building blocks in LSI SAG circuits.

### SAG multiplexers
We utilize the $V_{TH}$ reproducibility to operate the circuit shown in Fig. 2b. SAG FETs are connected in a hierarchical MUX structure, with each level consisting of devices fabricated on different rows of the SAG

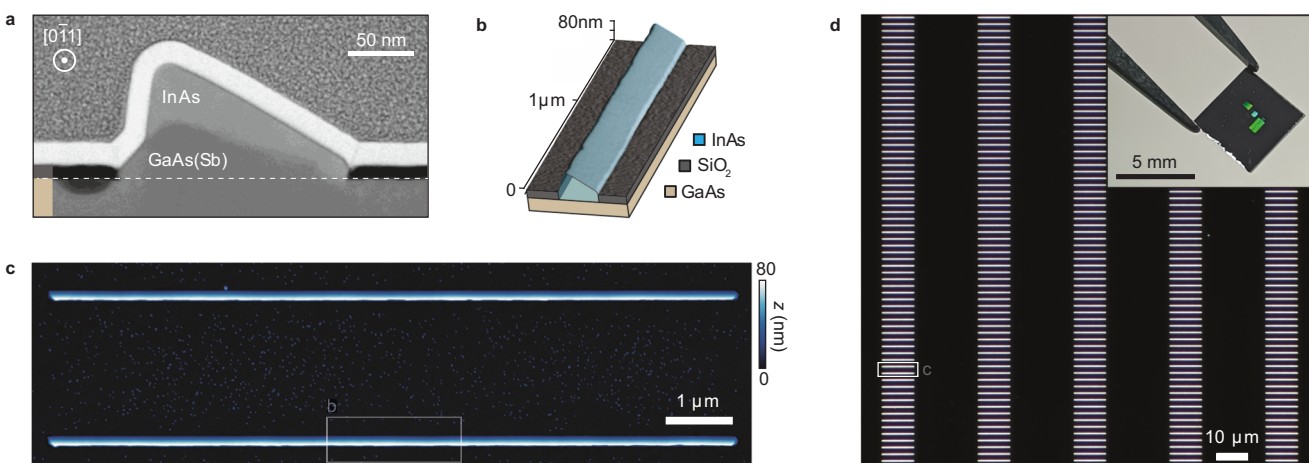

**Fig. 1 | Nanowire structure and morphology. a** Cross-sectional HAADF STEM micrograph of a SAG NW, showing the InAs conducting channel atop the GaAs substrate and GaAs(Sb) MBE-grown buffer. The NW exhibits an asymmetric triangular shape imposed by the $(3\,1\,1)$ substrate symmetry. The shape is not important for the present study which could equally well have been based on NWs grown on, e.g., $(1\,1\,1)$ or $(1\,0\,0)$ substrates with higher symmetry. **b** Combined schematic/3D AFM micrograph of a 2 μm long section of a single NW. **c** AFM micrograph of two SAG NWs. **d** Optical dark-field microscope image of a section of an InAs SAG NW array. Each NW is ~150 nm wide and 10 μm-long and individual NWs are spaced by $20 \times 2$ μm. Inset: Photograph of an as-grown sample. The large NW arrays are visible in green due to the diffraction of light.

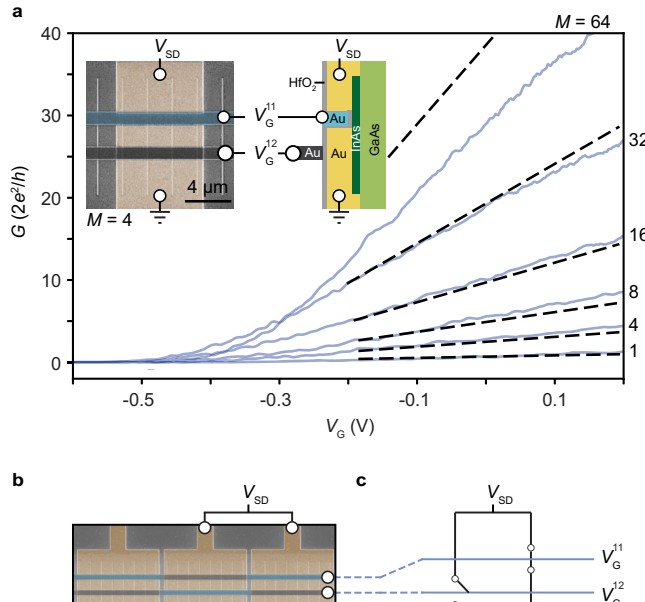

**Fig. 2 | Realizing a MUX circuit using NW FETs. a** Conductance, $G$, as a function of gate voltage $V_G$ for NW FETs based on 1, 4, 8, 16, 32, and 64 SAG NWs in parallel (blue) and expected linear trends (black dashed). The inset shows an SEM micrograph of an NW FET based on 4 SAG NWs, Ti/Au Ohmic contacts in gold, Ti/Au gate in blue. The gate in dark gray is screened by the underlying ohmic contact as illustrated by the cross-section schematic. **b** SEM micrograph of NW FETs arranged in a MUX circuit. Gates are screened in alternating elements as indicated by the blue/gray false coloring. **c** A schematic representation of the circuit in **b**.

NW array in Fig. 1d. Each gate spans the respective row, and the positions of the gated segment alternate such that for each NW FET, one gate (blue) tunes the carrier density of the NWs while the other is screened (gray). Input signals are thus directed through the MUX as illustrated in the schematic in Fig. 2c. With this design, each additional level doubles the number of outputs such that an $n$ level MUX has $2^n$ outputs and requires $2n$ gates for operation.

Figure 3a shows an optical micrograph of an 8-level MUX circuit connected back-to-back to a corresponding 8-level d-MUX. The circuit has a footprint of ~ 0.6 × 1.1 mm and incorporates 8192 individual SAG NWs in the form of 1996 interconnected FETs. Combining the 32 gate lines with two separate source-drain pairs enables individual addressing of any of the 512 devices under test (DUT) located in the gap between the MUX/d-MUX units. Where the DUT themselves consist of FETs with a single common gate, and 37 control lines thereby enabling experiments on 512 devices. In our case, the DUT in Fig. 3a consists of SAG devices with different functionalities and properties. For example, the SEM micrograph in Fig. 3b shows DUT devices #70–#77. Odd-numbered devices #71, #73, #75, #77 are SAG NW FETs with a contact separation of 100 nm and a common top gate. The even-numbered channels consist of continuous metal paths covering the NW. These allow confirmation of the MUX/d-MUX function irrespective of the DUT performance and also provide reference measurements of the MUX and d-MUX series resistance.

Before discussing DUT properties, we analyze the functionality of the MUX/d-MUX circuit. Figure 3c shows the conductance of the

circuit for each of the 65536 combinations of the first 256 MUX and d-MUX channels. The measurement was performed with positive voltage on the DUT gates, which were therefore all conducting. Indeed, high conductance is observed along the main diagonal, ($\alpha$), which corresponds to both MUX and d-MUX addressing the same DUT channel. This confirms that none of the 1996 SAG NW FETs of the MUX/d-MUX pair fail to conduct which would lead to regions of no conductance along the diagonal. In the case of negative $V_G$ on the DUT level, every second pixel of the diagonal has $G = 0$ (Supplementary section S5). In the ideal case, the diagonal would be the only non-zero value of the conductance matrix. However, finite off-diagonal conductivity also appears following a repeating pattern every 4, 64, and 128 channels in Fig. 3c and d ($\beta, \gamma, \delta$). Since the FETs are conducting at $V_G = 0$ V, finite current at these combinations of MUX and d-MUX channels corresponds to rows of NW FETs failing to respond to the gates. This was likely due to a break in gate lines or a failing bond wire, which can occur for large, complex circuits. Figure 3e, f schematically illustrates the correlation between the patterns of the matrix and FETs failing to pinch-off turn-off at various positions in the circuit. For example, a non-responsive gate, on the second d-MUX level from the DUT layer (blue cross in Fig. 3e) would allow transport for (MUX, d-MUX) combinations (2, 0), (3, 1), (6, 4), and (7, 5) as indicated by blue in Fig. 3f. Comparing to the measurement in Fig. 3c, d the periodically repeating off-diagonal pattern can be assigned to failures of one of the gates in the MUX levels marked with the corresponding labels in Fig. 3a. The additional feature appearing at d-MUX channel 192 (*) results from the combination of faulty FETs at channels 64 and 128 (Supplementary Section S7 and S8 provide further analysis of the faults of the circuit).

Importantly, the MUX/d-MUX configuration allows for identifying and in most cases self-corrects for malfunctioning elements of the SAG circuit; the double redundancy makes the measurement tolerant towards non-symmetrical errors, as, e.g., a non-functioning gate on the MUX side will be intrinsically corrected for by the function of the corresponding d-MUX gate. While errors appearing symmetrically in the MUX and d-MUX side of the circuit cannot be corrected for, they can be identified in the conductance matrix and the corresponding DUT can be excluded from experiments/analysis. This is schematically illustrated in Fig. 3e, f: if the MUX/d-MUX FETs fail to pinch-off at the symmetric red/purple positions, addressing levels 0, 1, 2, and 3 would also mix signals from levels 4, 5, 6, 7, respectively. Such a situation is readily identified by the symmetric off-diagonal nonvanishing elements in the conductance matrix (purple and red in Fig. 3f). We note that FETs failing to pinch-off would pass unnoticed in single-ended MUX layouts[32] where DUT share a common ground. The opposite case, where MUX FETs fail to open would result in periodic non-conductive elements in the diagonal of the matrix. Other examples of MUX/d-MUX circuits are discussed in Supplementary Section S6, showing that even with the number of failures typical for research-level devices, the self-correcting nature of MUX/d-MUX configuration generally protects against a reduction in the available number of DUT. We note that broken gate lines or failing bond wires should be readily eliminated and with a near unity yield of the NW FETs, future device generations could be successfully operated in a single-ended configuration.

As a final comment on MUX operation, we note that bandwidth is a key issue for control electronics. In our experiments, bandwidth was limited by the cryogenic setup, being optimized for low electron temperature, including 5 kHz low-pass filtering of each line. The MUX operation was uninhibited up to these frequencies (Supplementary Section S9). While we expect much higher bandwidth for individual NW similar to previous InAs NW devices operating at GHz[11,38,39], a radio-frequency MUX will require a circuit redesign to account, e.g., for impedance matching at every node, and may likely require additional fabrication steps. We also note, that while heat dissipation is negligible in the current experiments it could become relevant upon increasing

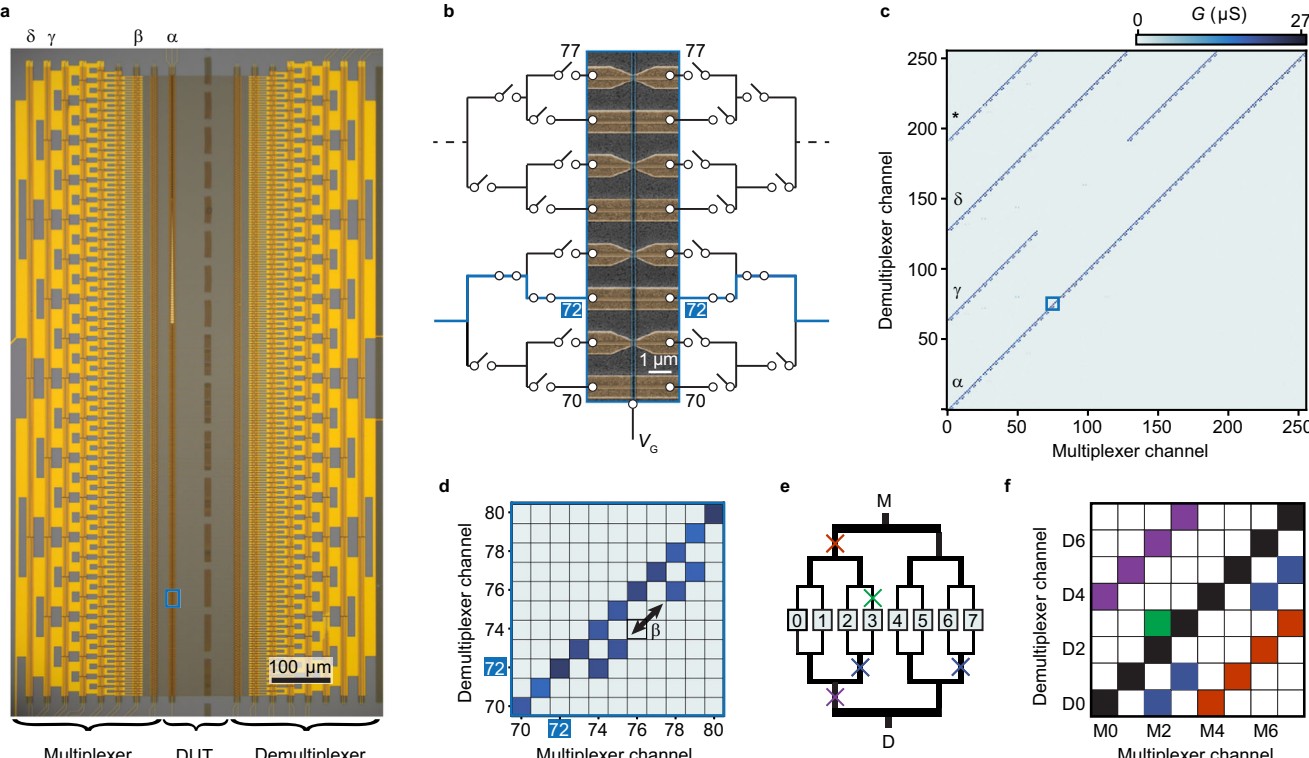

**Fig. 3 | Operation of a NW-based MUX-deMUX circuit. a** Optical microscope image of MUX/d-MUX circuit based on InAs SAG NW arrays. Devices under test (DUT) are labeled $\alpha$ and labels $\beta, \gamma, \delta$ indicate FET gates failing to deplete and related to corresponding off-diagonal signals in panel **c**. For the device shown 500 of the 512 lines were connected at the DUT level. The number of DUT was doubled from $2^8 = 256$ by using two source/drain pairs. **b** SEM micrograph of the DUT area indicated by the blue box in panel **a**, showing 8 devices connected to the MUX and d-MUX channels. Every second channel is shorted for use as a reference to obtain the series resistance of the adjacent channel. **c** Conductance matrix of all 65,536 combinations of source and drain channels of the first 256 connections of the circuit. The color of each pixel corresponds to the measured conductance of the specific channel combination. The uninterrupted diagonal feature shows that all of the DUTs are addressable and conducting. **d** Expanded view of the conductance matrix within the blue square in **c**. **e** Schematic three-level MUX/d-MUX. The colored crosses mark a FET failing to deplete and panel **f** shows the corresponding signatures on the conductance matrix. When operated at the diagonal, the circuit is immune to such errors except for symmetric pairs such as red/purple. This case can, however, be identified in the conductance matrix as regions with symmetric off-diagonal finite conductance, and accounted for in subsequent measurements.

operation frequency and/or further scaling up of the number of channels.

## Multiplexing of QD arrays

Eliminating the device-count roadblock by the MUX/d-MUX circuit enables fundamentally new experimental approaches in quantum electronics. For example, adding statistical significance to the characterization and optimization of device performance and material properties is crucial for efforts toward up-scaling quantum circuits. As an example, we demonstrate here the use of the circuit for establishing statistical reproducibility within large ensembles of lithographically identical devices. An array of 50 lithographically identical SAG QD devices were embedded in the DUT layer as shown in Fig. 4a. Potentials $(V_T, V_M, V_B)$ applied to three shared gates (top, middle, bottom) simultaneously tune the electrostatics of all 50 devices. This cross-bar approach is an important strategy for limiting the gate count in the up-scaling of QD arrays, however, the successful operation requires significant reproducibility between devices[36,37]. Here, we benchmark the consistency in the SAG NW QD array by comparing the statistical distributions of QD parameters among devices labeled Dev1-Dev20. First, however, since InAs SAG QDs have thus far not been demonstrated, we establish the characteristics of a single device (Dev1).

Figure 4c shows $G$ vs. $V_T$ and $V_B$ for fixed $V_M = 1$ V. Pinch-off is at $\sim -150$ mV for both $V_T$ and $V_B$, and horizontal/vertical structures are attributed to resonances below each gate modulating the transmission[5]. With both gates near pinch-off, electrons are ideally

confined to an NW segment below the middle gate, thus defining a QD. Indeed, fixing $V_T$ and $V_B$ at the position of the red dot in Fig. 4c, diamond-shaped regions of low conductance associated with Coulomb blockade (CB) are observed in the map of the differential conductance, $dI/dV_{SD}$, vs. source bias $V_{SD}$ and $V_M$, as shown in Fig. 4b. The $V_{SD}$-height of the diamonds provides an estimate of the QD addition energy, being the sum of the electrostatic charging energy $E_C$ and the single-particle level spacing $\Delta E$. As discussed in Supplementary Section S10 the QDs have $\Delta E \ll E_C$ and from Fig. 4b we estimate $E_C \sim 210 \pm 15$ µeV. The capacitance between the QD and the middle plunger gate is estimated as $C_G = e / \overline{\Delta V_M} = 0.4$ fF where $\overline{\Delta V_M} = 0.46$ mV is the average value CB peaks spacings in the range of Fig. 4b. This $C_G$ value agrees with the result (0.37fF) of simple capacitance estimate based on the gate layout (Supplementary Section S10) and thus supports that in this particular gate-configuration, the QD confinement is defined by gates as intended.

To investigate the sensitivity to the tuning, $G(V_M)$ was measured at $11 \times 11$ equally spaced $(V_T, V_B)$ points, spanning the white square in Fig. 4c. CB peaks were identified at 108 of the 121 gate-tunings and Fig. 4d shows the corresponding map of $\overline{\Delta V_M}$. No systematic trend is observed and the distribution of $\overline{\Delta V_M}$ shown in Fig. 4e (top), is symmetric with a mean $\overline{\overline{\Delta V_M}} = 464$ µV and standard deviation $\sigma = \pm 27$ µV, again consistent with a QD defined between the top and bottom gates.

The choice of range for the cross-bar gate-tunings in Fig. 4b, d was based on the gate characterization specific for Dev1 (Fig. 4c). We now use the MUX circuit to gather statistics for the different devices in

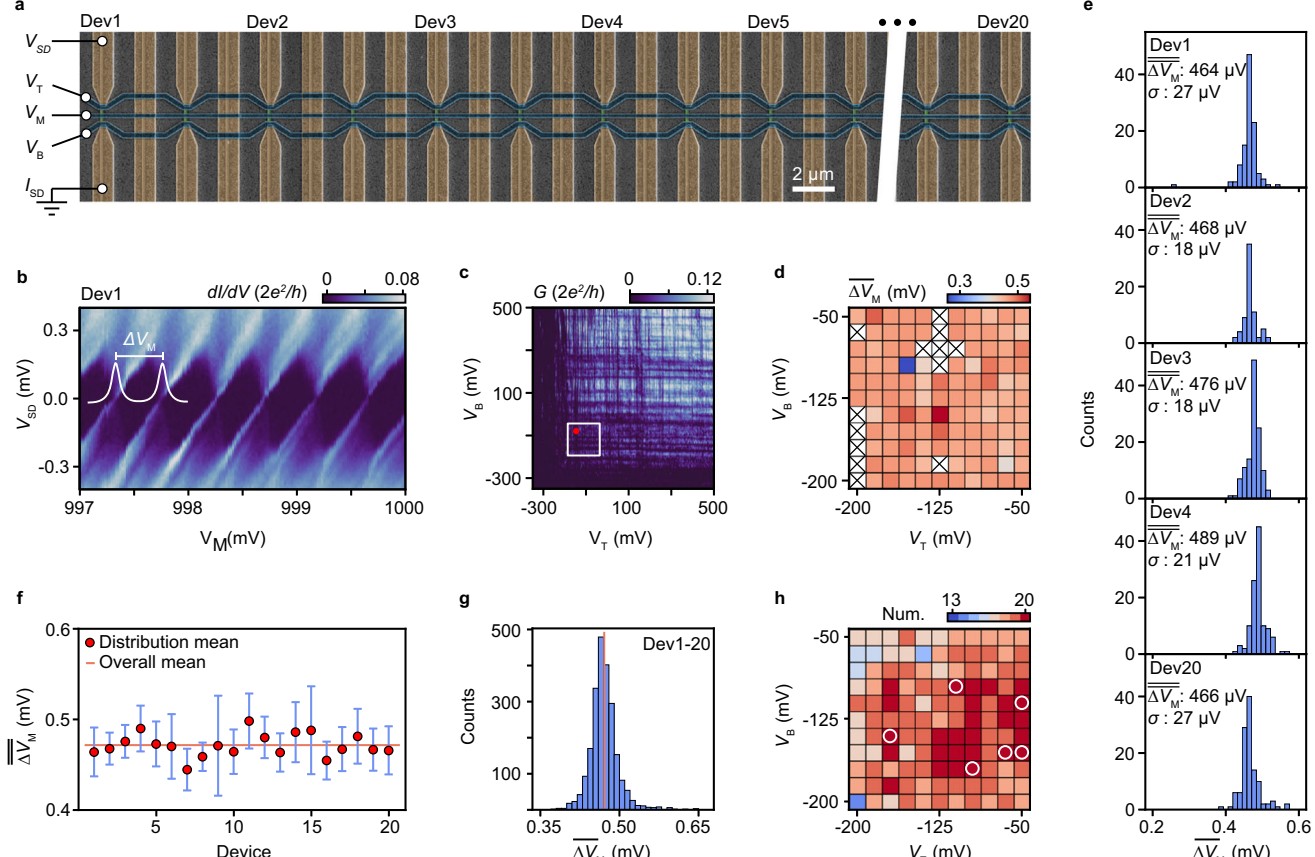

**Fig. 4 | Characterization of the SAG QD ensemble. a** SEM micrograph of an array of NW QD devices. Contacts (gold) are spaced by 900 nm, top/middle/bottom cross-bar gates (blue) are 300/200/300 nm wide and have a 150 nm spacing. The gates are shared between 50 devices of which 20 were analyzed in detail. Each device is paired with a short line for reference. **b** Source-drain bias spectrum (differential conductance $dI/dV$ vs. $V_{SD}$ and $V_M$) of Dev1 showing Coulomb diamonds. When sweeping $V_M$ the barrier gates were compensated for a slight capacitive cross coupling (Supplementary Section S10). $V_T$ and $V_B$ starting values correspond to the red point in **c**. **c** Conductance $G$ vs. $V_T$ and $V_B$ for Dev1. **d** Map showing $\overline{\Delta V_M}$ of Dev1 for various $V_T$ and $V_B$ combinations within the white square in **c**. Crossed points have no identifiable Coulomb peaks. **e** Histograms of $\overline{\Delta V_M}$ for Dev1-Dev4 and Dev20 in the same $V_T$ and $V_B$ range. **f** Distribution of $\overline{\Delta V_M}$ for Dev1-Dev20. The red dots indicate $\overline{\Delta V_M}$ of each device and orange line shows $\overline{\overline{\Delta V_M}}$ across all devices, error bars show the standard deviation. **g** Histogram showing overall distribution of $\overline{\Delta V_M}$ for all devices. Orange line indicates the average value and corresponds to the line in **f**. **h** Number of devices with observable Coulomb peaks at each combination of $V_T$ and $V_B$. White circles show points where all 20 devices have $\overline{\Delta V_M}$ within $2\sigma$ of the common average.

order to probe the consistency across the array while keeping these same tuning parameters. Values of $\overline{\Delta V_M}$ were extracted from $G(V_M)$ traces measured for all devices and all gate-points. All devices were operational and exhibited coulomb blockade and from the resulting 2420 gate-traces, 17,924 CB peaks were identified and fitted. Examples of measured data and peak analysis are presented in Supplementary Sections S12 and S13. The distributions of $\overline{\Delta V_M}$ for all devices are included in Supplementary Section S14 and examples for Dev1-4 and Dev20 are shown in Fig. 4e. A comparison of the distributions and their mean values, $\overline{\Delta V_M}$, among the devices of the array, is shown in Fig. 4f. Except for Dev7, the distribution means fall within one $\sigma$ of the overall common mean. The spread between devices could be affected by structural variations between nanowires due to SAG processing or to variations in post-growth device processing. The spread within each devices could be related to changes in the effective confinement potential with gate tunings and may be different between devices due to random impurity in the vicinity of the devices.

Finally, Fig. 4g shows the joint distribution of all $\overline{\Delta V_M}$ and Fig. 4h illustrates the number of devices displaying CB for all measured combinations of $V_T$, $V_B$. In 27 out of the 121 point in cross-bar gate space, all 20 devices simultaneously exhibited CB. The circles mark the 7 tunings where all devices fulfill the stricter criterion of showing CB and having $\overline{\Delta V_M}$ within $\pm 2\sigma$ of the joint mean peak spacing.

Figure 4f–h constitutes a key result of the current study, establishing both a level of device-to-device reproducibility supporting the potential of SAG for as a scalable platform for quantum electronics. Further, the statistical bench-marking of the QD devices explicitly demonstrates a key example of the new possibilities enabled by the integration of MUX/d-MUX circuits.

## Discussion

In conclusion, we successfully fabricated and operated cryogenic multiplexers/de-multiplexer circuits based on InAs NWs grown bottom-up by selective area growth. The circuit removes the limitations on device count in conventional cryogenic electronics thus enabling new experimental strategies such as searches through large ensembles of devices for rare or exotic phenomena, establishing the correlation between device performance and materials properties or device geometry, and establishing the statistical reproducibility among devices—a prerequisite for further scaling quantum of circuits. This capacity was demonstrated by statistically characterizing an ensemble of SAG QDs. The methods developed here complement cryo complementary metal oxide semiconductor technologies by enabling scaling and integration within bottom-up quantum materials which constitute a unique platform, e.g., with hybrid semiconductor/superconductor quantum technologies. In general, the methods

enable optimization of quantum materials and devices based on the automated acquisition of statistically significant datasets rather than proof-of-principle examples. This direction will be empowered by the ongoing developments of advanced data evaluation[40] and machine learning[41–43] for unsupervised and optimized acquisition and tuning of large ensembles of quantum devices with many tuning parameters[44–46]. The circuit may be expanded further by replacing the single lines in the current design with a multi-channel bus[47] to enable, e.g., integration of charge sensors, multi-terminal devices, complex gate architectures, and/or the operating the MUX as a multi-channel DAC[48].

## Methods

For SAG fabrication and synthesis, a 10 nm $SiO_2$ mask layer was first deposited on epi-ready GaAs (3 1 1)A substrates by plasma-enhanced chemical vapor deposition. $0.15 \times 10\,\mu m$ rectangular openings were defined in the oxide along the [0 $\bar{1}$ 1] direction by e-beam lithography (EBL) and dry etching. The openings were arranged in $512 \times 16$ arrays with a pitch of 2 $\mu m$ and 20 $\mu m$ along the [0 $\bar{1}$ 1] and [$\bar{2}$ 3 3], respectively (Fig. 1d). GaAs(Sb)/InAs double layer NWs were selectively grown in the openings where the GaAs(Sb) buffer was introduced to improve the crystal surface for the subsequent InAs transport channel[20]. Synthesis details and structural analysis are provided in Supplementary Sections S1 and S2. For device fabrication, Ti/Au ohmic contacts to the SAG NWs were defined on the growth substrate by standard EBL, metal evaporation, and liftoff. Subsequently, a 15 nm $HfO_2$ gate dielectric was deposited by atomic layer deposition and top-gates were defined by electron beam lithography, metal evaporation, and liftoff. The QD devices in Fig. 4 have a contact separation of 900 nm, top/middle/bottom cross-bar gates are 300/200/300 nm wide, and have a 150 nm spacing. Electrical measurements were carried out in a dilution refrigerator with a base temperature of 20 mK. The conductance, $G = I/V_{SD}$, where $I$ is the drain current generated by source voltage $V_{SD}$, was measured as a function of the gate potential, $V_G$, using standard lock-in techniques. The series resistance $R_S$ for data presented in Fig. 2 was estimated by fitting the $G(V_G)$ traces with the standard expression[49] $G = (R_S + \frac{L^2}{\mu_{FE}C(V_G - V_{TH})})^{-1}$ where $L$ is the gate length, $C$ is gate capacitance simulated as described in Supplementary Sections S3 and S4, and $\mu_{FE}$ is the electron mobility. When sweeping $V_M$ in the QD measurements, the barrier gates were compensated for a slight capacitive cross-coupling (Supplementary Section S10).

## Data availability

The electrical transport data generated in this study have been deposited in the Figshare database under the accession code https://doi.org/10.11583/DTU.c.6788313.

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

## Acknowledgements

This research was supported by research grants from Villum Fonden (Grant no.: 00013157) (T.S.J.) and the European Research Council under the European Union's Horizon 2020 research and innovation program (Grant no.: 716655 and 866158) (T.S.J.) and Microsoft Quantum. ICN2 acknowledges funding from Generalitat de Catalunya (Grant no.: 2021SGR00457) (J.A.). This study is part of the Advanced Materials program and was supported by MCIN with funding from the European Union NextGenerationEU (Grant no.: PRTR-C17.I1) (J.A.) and by General-itat de Catalunya. The authors thank the support from "ERDF A Way of Making Europe", by the "European Union". ICN2 is supported by the Severo Ochoa program from Spanish MCIN/AEI (Grant no.: CEX2021-001214-S) and is funded by the CERCA Program/Generalitat de Catalunya. Authors acknowledge the use of instrumentation as well as the technical advice provided by the National Facility ELECMI ICTS, node "Laboratorio de Microscopías Avanzadas" at the University of Zaragoza. We acknowledge support from CSIC Interdisciplinary Thematic Platform (PTI+) on Quantum Technologies (PTI-QTEP+).

## Author contributions

D.O. fabricated devices, performed electrical measurements, and analyzed data. G.N. developed NW growth and grew NW samples, performed AFM, and analyzed data. D.C. performed electrical measurements and analyzed data. D.B. developed NW growth. C.P. analyzed data. S.M.S. and J.A. performed cross-section TEM, GPA, and EELS analysis. T.S.J. conceptualized the experiment and analyzed the data.

## Competing interests

The authors declare no competing interests.
