## [Peer Review File · Nature Communications]

REVIEWER COMMENTS

Reviewer #1 (Remarks to the Author):

In their work 'Cryogenic Multiplexing using Selective Area Grown Nanowires', D. Olsteins et al. demonstrate very significant progress in scaling up bottom-up III-V selective area nanowire devices, with remarkable performance, novelty, and a thorough statistical analysis of all results. The work is overall of the utmost scientific rigor and quality. The novel results presented in the main manuscript are solid and backed by an excellent set of 14 complementary supporting figures/analyses.

D. Olsteins et al. first introduce a MUX/d-MUX addressing scheme to be able to address and measure 512 individual InAs SAG FET devices with 37 control lines (out of 1996 fabricated interconnected FETs), therefore enabling the automated acquisition of truly statistically significant datasets. 4 types of these devices are shown in total, incl. in the supporting information file, demonstrating robustness and reproducible fabrication capability. The consistency and reproducibility of the devices as well as the ability to identify and sometimes self-correct the faulty ones directly result from the newly demonstrated ability in this work to address and control so many devices. It is the first time authors have come up with a means to study the reproducibility and performance of bottom-up grown devices on this parallel, massive scale.

Further, the authors demonstrate the tuning, operation, and statistical characterization of InAs SAG quantum dots, which have a relatively low device-to-device variation. Importantly, the authors are the first to report on this type of device in the SAG material system, as it had not even been shown for a single InAs/GaAs(Sb) nanowire before. The authors can simultaneously tune the electrostatics of all 50 devices using 3 shared gates. The methodology presented in this work can readily be extended to a very large number of systems and answer many physics questions as well as enable for the first time strong and fast feedback look on the device fabrication/structural characterization/growth variation interactions, which is strictly necessary as a first step toward scaling up bottom-up quantum electronic devices.

I believe such an advance in the field was absolutely needed and therefore very timely. It brings a positive step in the direction of being able to integrate the advantages of bottom-up grown advanced quantum materials with the need for scaled-up high-speed capable quantum electronics, operating at 20mK. The outlooks about future device architectures and the use of advanced data evaluation and machine learning make sense.

I suggest this manuscript be accepted as is.

Minor points: two typos in the Supp. Information: p13, figure caption b 'Illustratino'; p17, S14 title 'Histograms'. I would suggest the authors add an 'author contribution statement'.

Reviewer #2 (Remarks to the Author):

This is a carefully written paper reporting multiplexing of nanowires devices, in which the nanowires are grown by selective area growth, and the multiplexer and demultiplexer are also formed from the same type of nanowires. The results are well-described and clear.

This paper will make clear contributions to several different fields: selective area growth at a high level, integration of many such device, cryogenic operation of highly integrated devices, characterization in parallel of many such devices. The advances in each area are reasonable. Taken together, the paper is quite strong indeed. This paper is a strong example of cryogenic semiconductor chips taking a substantial step up in complexity, which is important for technology development. The authors make clear that the device is not perfect.

This leaves the question of the title and abstract. They are both clearly correct as written, and we could stop there. At the same time, they do not really address what is most interesting, which is in some sense the complex integration of devices that function well at low temperature and display simple but real quantum effects. Having said that, I cannot think of a short title that is better. Thus, this is not really a suggestion or a complaint, but rather a comment that more is done here than meets the eye at first glance at the title and abstract.

Reviewer #3 (Remarks to the Author):

The manuscript entitled "Cryogenic multiplexing using selective area grown nanowires" by authors D. Olsteins et al. describes cryogenic multiplexing and de-multiplexing using selective area grown nanowires in the dc domain. It is impressive to see that the authors managed to fabricate and control 8192 nanowires wire relatively high yield, which is functionally further improved due to redundancy in the individual control for the mux and the de-mux. The authors also measured 50 SAGNW quantum dots and collected statistics of their CB peak spacing ΔV_m for different devices and operating regimes. The manuscript is well written and the results are in principle interesting for a wider audience. I would recommend publication in Nat. Commun, provided the following comments are addressed.

1) Looking from a larger perspective, similar multiplexing and demultiplexing schemes have been demonstrated before in different technologies with much higher, typically unit yield, e.g. in cryoCMOS [29]. Apart from on-chip co-integration of the mux and the de-mux with nanowire quantum dots are there any other advantages of this technology, such as small size, large dynamic range, good thermalization, low leakage, etc? A short paragraph discussing this technologies advantages would be welcomed.

2) In cryoCMOS and other technologies, heat dissipation is carefully monitored and minimized especially at millikelvin temperatures where cooling power is limited or when operating quantum devices that are heavily susceptible to electromagnetic noise, such as superconducting qubits. Have the authors characterized potential local heating sources such as gate to NW leakage? I understand that in small device number experiments with quantum dots leakage is typically negligible, however, this manuscript discusses possibility of LSI and demonstrates control of thousands of NWs, with a potential to scale to even larger numbers. In this case the leakage could accumulate into a detectable amount.

3) The authors carefully discuss pinch-off failures that lead to off-diagonal conductivity and claim that these originate from broken wire bonds or defects in gate-lines, while the yield of NWs themselves seems 100% since all diagonal elements (α) are conducting and there are no symmetric off-diagonal features in the conductance matrix. I find it strange that most of the failures happen on much larger structures (gate lines) and not on the very thin nanowires. Have the authors characterize the performance of nanowires independently to determine their fabrication yield, e.g. using critical dimension scanning electron microscope CDSEM? To me, failed wirebonds or defective wide control lines do not seem to represent a fundamental limitation to this technology's yield. So far the authors included only a discussion on how to detect or correct failures in double-ended mux/demux layouts which would not be possible in single-ended case, however, a comment on perspective of this technology with fundamental fabrication and yield limits should be added. In this way a better comparisons could be made to other multiplexing technologies.

4) At the end of the section SAG multiplexers, the authors discuss a possibility to extend the operating frequency to GHz range. While this is possible for individual nanowire devices or transistor circuits, a multiplexer that can efficiently operate in the microwave domain with high channel-to-channel isolation and low insertion loss would require a complete circuit redesign to ensure impedance matching at each node. Furthermore, any failure in control lines would create open parallel signal paths which would lead to frequency dependent impedance variation or standing waves in the through signal. To sufficiently reduce crosstalk (>30 dB) circuit features such as small size and possibility of LSI would likely be compromised, unless a more complex multilayer integration would be implemented that would enable signal fencing. For this reason I would recommend adding "low frequency" or "dc" to the title and revise the comment on high-frequency operation.

REVIEWER COMMENTS

Reviewer #1 (Remarks to the Author):

In their work ‘Cryogenic Multiplexing using Selective Area Grown Nanowires’, D. Olsteins et al. demonstrate very significant progress in scaling up bottom-up III-V selective area nanowire devices, with remarkable performance, novelty, and a thorough statistical analysis of all results. **The work is overall of the utmost scientific rigor and quality.** The novel results presented in the main manuscript are solid and backed by an excellent set of 14 complementary supporting figures/analyses.

D. Olsteins et al. first introduce a MUX/d-MUX addressing scheme to be able to address and measure 512 individual InAs SAG FET devices with 37 control lines (out of 1996 fabricated interconnected FETs), therefore enabling the automated acquisition of truly statistically significant datasets. 4 types of these devices are shown in total, incl. in the supporting information file, demonstrating robustness and reproducible fabrication capability. The consistency and reproducibility of the devices as well as the ability to identify and sometimes self-correct the faulty ones directly result from the newly demonstrated ability in this work to address and control so many devices. It is the first time authors have come up with a means to study the reproducibility and performance of bottom-up grown devices on this parallel, massive scale.

Further, the authors demonstrate the tuning, operation, and statistical characterization of InAs SAG quantum dots, which have a relatively low device-to-device variation. Importantly, the authors are the first to report on this type of device in the SAG material system, as it had not even been shown for a single InAs/GaAs(Sb) nanowire before. The authors can simultaneously tune the electrostatics of all 50 devices using 3 shared gates. The methodology presented in this work can readily be extended to a very large number of systems and answer many physics questions as well as enable for the first time strong and fast feedback look on the device fabrication/structural characterization/growth variation interactions, which is strictly necessary as a first step toward scaling up bottom-up quantum electronic devices. I believe such an advance in the field was absolutely needed and therefore very timely. It brings a positive step in the direction of being able to integrate the advantages of bottom-up grown advanced quantum materials with the need for scaled-up high-speed capable quantum electronics, operating at 20mK. The outlooks about future device architectures and the use of advanced data evaluation and machine learning make sense.

I suggest this manuscript be accepted as is.

Minor points: two typos in the Supp. Information: p13, figure caption b ‘Illustratino’; p17, S14 title ‘Histograms’. I would suggest the authors add an ‘author contribution statement’.

We thank the referee for careful reading of our manuscript and for the very positive assessment. We have fixed the typos in the supplementary and added an author contribution statement.

Reviewer #2 (Remarks to the Author):

This is a carefully written paper reporting multiplexing of nanowires devices, in which the nanowires are grown by selective area growth, and the multiplexer and demultiplexer are also formed from the same type of nanowires. The results are well-described and clear.

This paper will make clear contributions to several different fields: selective area growth at a high level, integration of many such device, cryogenic operation of highly integrated devices, characterization in parallel of many such devices. The advances in each area are reasonable. Taken together, the paper is quite strong indeed. This paper is a strong example of cryogenic semiconductor chips taking a substantial step up in complexity, which is important for technology development. The authors make clear that the device is not perfect.

This leaves the question of the title and abstract. They are both clearly correct as written, and we could stop there. At the same time, they do not really address what is most interesting, which is in some sense the complex integration of devices that function well at low temperature and display simple but real quantum effects. Having said that, I cannot think of a short title that is better. Thus, this is not really a suggestion or a complaint, but rather a comment that more is done here than meets the eye at first glance at the title and abstract.

We thank the referee for careful reading of our manuscript and for the very positive assessment. As the referee suggests we have modified the abstract to better cover the full breadth of our results. The abstract now reads:

Bottom-up grown nanomaterials play an integral role in the development of quantum technologies but are often challenging to characterise on large scales. Here, we harness selective area growth (SAG) of semiconductor nanowires (NWs) to demonstrate large scale integrated circuits and characterisation of large numbers of quantum devices. The circuit consisted of 512 quantum devices embedded within multiplexer/demultiplexer pairs, incorporating thousands of interconnected SAG NWs operating under deep cryogenic conditions. Multiplexers enable a range of new strategies in quantum device research and scaling by increasing the device count while limiting the number of connections between room-temperature control electronics and the cryogenic samples. As an example of this potential we perform a statistical

characterization of large arrays of identical SAG quantum dots thus establishing the feasibility of applying cross-bar gating strategies for efficient scaling of future SAG quantum circuits. More broadly, the ability to systematically characterize large numbers of devices provides new levels of statistical certainty to materials/device development.

Reviewer #3 (Remarks to the Author):

The manuscript entitled "Cryogenic multiplying using selective area grown nanowires" by authors D. Olsteins et al. describes cryogenic multiplexing and de-multiplexing using selective area grown nanowires in the dc domain. It is impressive to see that the authors managed to fabricate and control 8192 nanowires wire relatively high yield, which is functionally further improved due to redundancy in the individual control for the mux and the de-mux. The authors also measured 50 SAGNW quantum dots and collected statistics of their CB peak spacing ΔV_m for different devices and operating regimes. The manuscript is well written and the results are in principle interesting for a wider audience. I would recommend publication in Nat. Commun, provided the following comments are addressed.

We thank the referee for careful reading of our manuscript and for the positive assessment. Below we answer point-by-point to the questions of the referee.

1) Looking from a larger perspective, similar multiplexing and demultiplexing schemes have been demonstrated before in different technologies with much higher, typically unit yield, e.g. in cryoCMOS [29]. Apart from on-chip co-integration of the mux and the de-mux with nanowire quantum dots are there any other advantages of this technology, such as small size, large dynamic range, good thermalization, low leakage, etc? A short paragraph discussing this technologies advantages would be welcomed.

Large scale cryogenic circuits have indeed been demonstrated using cryoCMOS and also in AlGaAs based 2DEGs. While the SAG material platform cannot compete with cryoCMOS for speed or yield, bottom up III/V nanowires continues to be a workhorse for cryogenic quantum device research primarily. The reason is a combination of properties as discussed in the introduction to the manuscript where the following is stated: *Semiconductor nanowires (NWs) constitute an important platform for quantum electronics, since the electronic confinement intrinsic to the structure simplifies fabrication of complex devices[1-4], the flexibility of contact materials enables hybridization with important quantum materials such as superconductors[1,3,5-9], and an increased capacity for strain relaxation over bulk materials enables exploration of exotic heterostructures[2,6,8,10]].* Thus the importance of integration of the multiplexer with bottom-up SAG is not limited to QD research, but is relevant for the wide range of NW based research. As stated on p1. *"Introducing on-chip multiplexing to bottom-up grown nanostructures enables new strategies in quantum electronics research, such as automated searches through large ensembles of devices for rare or exotic phenomena, and systematic, statistically significant exploration of the correlation between device performance and e.g., materials properties or device geometry."* To make this point more clear - in light of the performance of cryoCMOS - we have added the following to the conclusion section:

Changing from: *This capacity was demonstrated by statistically characterizing an ensemble of SAG quantum dots. In general, the methods developed here enable optimization of quantum materials and devices based on automated acquisition of statistically significant datasets rather than proof-of-principle examples*

To: *This capacity was demonstrated by statistically characterizing an ensemble of SAG quantum dots. The methods developed here complement cryoCMOS technologies by enabling scaling and integration within bottom-up quantum materials which constitute a unique platform e.g. with hybrid semiconductor/superconductor quantum technologies. In general, the methods enable optimization of quantum materials and devices based on automated acquisition of statistically significant datasets rather than proof-of-principle examples.*

2) In cryoCMOS and other technologies, heat dissipation is carefully monitored and minimized especially at millikelvin temperatures where cooling power is limited or when operating quantum devices that are heavily susceptible to electromagnetic noise, such as superconducting qubits. Have the authors characterized potential local heating sources such as gate to NW leakage? I understand that in small device number experiments with quantum dots leakage is typically negligible, however, this manuscript discusses possibility of LSI and demonstrates control of thousands of NWs, with a potential to scale to even larger numbers. In this case the leakage could accumulate into a detectable amount.

Within the current experiments, operating roughly 10.000 wires at low frequencies the gate leakage or other effects provide negligible dissipation, and we observe no impact on the temperature of the system. Thus, no systematic investigation of dissipation was necessary. We agree, however, that with the perspective of future scaling and/or operation at high frequencies this should be considered. We have included a sentence mentioning this - please see

changes below, in our response to the question of high-frequency operation.

3) The authors carefully discuss pinch-off failures that lead to off-diagonal conductivity and claim that these originate from broken wire bonds or defects in gate-lines, while the yield of NWs themselves seems 100% since all diagonal elements (α) are conducting and there are no symmetric off-diagonal features in the conductance matrix. I find it strange that most of the failures happen on much larger structures (gate lines) and not on the very thin nanowires. Have the authors characterize the performance of nanowires independently to determine their fabrication yield, e.g. using critical dimension scanning electron microscope CDSEM? To me, failed wirebonds or defective wide control lines do not seem to represent a fundamental limitation to this technology's yield. So far the authors included only a discussion on how to detect or correct failures in double-ended mux/demux layouts which would not be possible in single-ended case, however, a comment on perspective of this technology with fundamental fabrication and yield limits should be added. In this way a better comparisons could be made to other multiplexing technologies.

We agree with the referee that the unity yield of the nanowire devices was a positive surprise. Nevertheless this is the case, and this is also consistent with a high structural uniformity of the NWs across the arrays as found by SEM and AFM. It is important to note that a lot of work by others has preceded these results - both pre-growth fabrication and ohmic contact recipes have been optimized for years. Coupled with high-end equipment, a 100% yield is not unlikely for devices with critical dimensions > 100 nm. Also, we agree with the referee that the failing bond wires is certainly not a fundamental limitation and would not be an issue in labs with high-end bonding capabilities. Thus there appears to be no fundamental effects limiting the yield and as requested by the referee we now explicitly state this in the manuscript by adding the following sentence: *We note that broken gate-lines or failing bond-wires should be readily eliminated and with a near unity yield of the NWFETs, future device generations could be successfully operated in a single-ended configuration.* This is added after the sentence: *Other examples of MUX/d-MUX circuits are discussed in Supplementary Section S6, showing that even with the amount of failures typical for research-level devices, the self-correcting nature of MUX/d-MUX configuration generally protects against a reduction in the available number of DUT.* on page 4.

4) At the end of the section SAG multiplexers, the authors discuss a possibility to extend the operating frequency to GHz range. While this is possible for individual nanowire devices or transistor circuits, a multiplexer that can efficiently operate in the microwave domain with high channel-to-channel isolation and low insertion loss would require a complete circuit redesign to ensure impedance matching at each node. Furthermore, any failure in control lines would create open parallel signal paths which would lead to frequency dependent impedance variation or standing waves in the through signal. To sufficiently reduce crosstalk (≥ 30 dB) circuit features such as small size and possibility of LSI would likely be compromised, unless a more complex multilayer integration would be implemented that would enable signal fencing. For this reason I would recommend adding "low frequency" or "dc" to the title and revise the comment on high-frequency operation.

We thank the referee for this insightful comment. We acknowledge that although single NWs may operate at microwave frequencies the design and operation of complex circuit can be much more involved. We have now included these reservations the paragraph concerning the bandwidth. It was changed from:

As a final comment on MUX operation, we note that bandwidth is a key issue for control electronics. In our experiments, bandwidth was limited by the cryogenic setup, being optimized for low electron temperature, including 5 kHz low-pass filtering of each line. The MUX operation was uninhibited up to these frequencies (Supplementary Section S9), and we expect much higher bandwidth to be possible similar to other InAs NW electronics operating at GHz[11,38,39]

To: *As a final comment on MUX operation, we note that bandwidth is a key issue for control electronics. In our experiments, bandwidth was limited by the cryogenic setup, being optimized for low electron temperature, including 5 kHz low-pass filtering of each line. The MUX operation was uninhibited up to these frequencies (Supplementary Section S9). While we expect much higher bandwidth for individual NW similar to previous InAs NW devices operating at GHz[11,38,39], a radio-frequency MUX will require a circuit redesign to account e.g. for impedance matching at every node and may likely require additional fabrication steps. We also note, that while heat dissipation is negligible in the current experiments it could become relevant upon increasing operation frequency and/or further scaling of the number of channels.*

Also, since this is the first example of multiplexing with bottom up nanowires, and we did not at this point explore the limits of increasing operation frequency, we do not wish to limit the scope of the manuscript by adding "low

frequency” to the title. This is also supported by reviewer 2 suggesting maybe an even broader title. After careful consideration we prefer to keep the title as is.

REVIEWERS' COMMENTS

Reviewer #1 (Remarks to the Author):

The authors have taken into account my minor comments and fully implemented the suggested corrections. I have also read the comments from the other two reviewers and checked the revisions to the manuscript. As a result, the article is even clearer in its revised form.

The work is important and timely, with impact for a broad community ranging from quantum transport to semiconductor nanotechnology characterization and growth.

In my opinion, the article is ready to be published in its current form.

Reviewer #2 (Remarks to the Author):

My only previous concern has been addressed, namely that the work did more than was really advertised. I recommend publication.